

# Sea ice reduction in the Barents-Kara Sea enhances June precipitation in the Yangtze River basin

Tianli Xie[1], Zhen-Qiang Zhou[1, 2, *], Renhe Zhang[1, 2], Bingyi Wu[1, 2], Peng Zhang[1, 2]

[1] Department of Atmospheric and Oceanic Sciences, Institute of Atmospheric Sciences, Fudan University, Shanghai, China

[2] CMA-FDU Joint Laboratory of Marine Meteorology and Key Laboratory of Polar Atmosphere-Ocean-Ice System for Weather and Climate, Shanghai, China

*Correspondence to*: Zhen-Qiang Zhou, zqzhou@fudan.edu.cn

**Abstract.** This study investigates the influence of June sea surface temperature (SST) and sea ice in the Barents-Kara Sea (BKS) on concurrent rainfall variability in the Yangtze River basin from 1982 to 2021 using both observational data and numerical experiments. The observed decrease in BKS sea ice and the corresponding increase in SST during June aligns with enhanced precipitation in the Yangtze River basin on the interannual timescale. The BKS thermal forcing induces an equivalent barotropic Rossby wave train in the middle and upper troposphere, which propagates southeastward to the Northwest Pacific (NWP). This Rossby wave train features two positive centers over the BKS and NWP, and one negative center above the Baikal Lake. The strengthened NWP subtropical high and upper-level westerly jet contribute to increased rainfall in the Yangtze River basin by enhancing moisture transport and anomalous ascending motions. These findings provide important implications for predicting summer rainfall in East Asia.

## 1 Introduction

East Asia, characterized by its dense population and swift economic growth, experiences a precipitation belt during the early summer that stretches from the Yangtze River basin in China to the central-southern regions of Japan and the southern part of South Korea. This rain belt, known as the Mei-



yu rainband in China, exhibits a quasi-stationary state with pronounced year-to-year variability. Fluctuations in precipitation directly affect local water resources, agricultural policies, disaster prevention,

and various socioeconomic factors (Ninomiya and Murakami, 1987; Fu and Teng, 1988; Xu et al., 2000; Huang et al., 2004; Ding et al., 2020).

Sea surface temperature (SST) variability in the tropics is an important driver for interannual variability in summer rainfall over the East Asia region (Xie et al., 2016; Zhang et al., 2017). The El Niño-Southern Oscillation (ENSO) in the Pacific Ocean has been identified as the dominant forcing of

Mei-yu rainfall variability (Huang and Wu, 1989; Huang et al., 2004; Xie et al., 2016; Zhang et al., 2017; Ding et al., 2020). During post-El Niño summers, Mei-yu in East Asia tends to increase with a low-level Northwest Pacific (NWP) anomalous anticyclone (AAC). The NWP AAC develops rapidly during an El Niño winter (Zhang et al., 1996) and lasts through the following summer through many mechanisms (Huang and Zhang, 1997; Wang et al., 2000; Xie et al., 2009; Rong et al., 2010). However, recent studies

found that the excessive Mei-yu rainfall and NWP AAC are often but not always induced by ENSO, such as in the year of 2020. For the 2020 summer, the Mei-yu rainfall in the Yangtze River basin surpassed historical records dating back to 1961, causing severe flooding, while the 2019-20 El Niño event was very weak. Indeed, the extreme Mei-yu event of 2020 is largely due to the extreme Indian Ocean dipole (IOD) event of 2019 (Takaya et al., 2020; Zhou et al., 2021). The Indian Ocean warming in summer 2020

induced the NWP AAC and intensified the upper-level westerly jet over the Yangtze River basin, causing heavy rainfall along the Yangtze River (Zhou et al., 2021). Positive SST anomalies in May over the North Atlantic can also enhance rainfall in June over the Yangtze River basin (Zheng and Wang, 2021).

In addition to tropical SST forcings, sea ice variations in the Arctic have increasingly drawn attention to their impact on climate in the Eurasian region since the 20th century (Gao et al., 2015; Wu et al., 2017;

Hou et al., 2022). Since the 20th century, there have been rapid changes in both SST and sea ice in the Arctic (Comiso, 2006; Francis and Hunter, 2007; Stroeve et al., 2007; Screen and Simmonds, 2010). The Arctic sea ice variations could influence the East Asian summer monsoon by modulating the wave train along the Asian jet. The declining sea ice in the Arctic during spring and summer leads to an enhancement of summer rainfall in northeast China (Niu et al., 2003; Zhang et al., 2021), central China between the

Yangtze River and the Yellow River (Wu et al., 2009; He et al., 2018; Zhang et al., 2018; Shen et al.,



2019; Chen et al., 2021), Indochina Peninsula, and the Philippines and decreased rainfall over Mei-yu front zone (Guo et al., 2014). For instance, He et al. (2018) demonstrated that June sea ice variability in the Barents Sea can trigger a meridional wave train extending to midlatitudes, which further influences the Silk Road pattern and consequently impacts East Asian precipitation in August. Wu et al. (2009)

found that variations in spring sea ice concentrations in the Arctic Ocean and the Greenland Sea can trigger a Eurasian wave train that extends from northern Europe to Northeast Asia, thereby influencing summer rainfall patterns in China.

However, there is ongoing debate regarding whether changes in Arctic sea ice can influence midlatitude climate and atmospheric circulation (Cohen et al., 2020). A recent study found that the

decrease in Arctic sea ice has little impact on summer mean precipitation in the Yangtze River basin and South China, with its main impact being observed in the mid-latitude and high-latitude regions of East Asia (Wu et al., 2023). Over the past decades, there has been a substantial reduction in the extent of summer sea ice in the Arctic (Onarheim et al., 2018; Kumar et al., 2021). The Barents-Kara Sea (BKS), which is recognized for experiencing one of the most rapid rates of sea ice decline in the Arctic,

demonstrates significantly elevated SSTs during the summer months compared to the average (Carvalho and Wang, 2020; Yang et al., 2023). Nevertheless, the simultaneous impact of sea ice and SST variations in BKS on summer rainfall in the Yangtze River basin, remains unclear and contentious. The present study investigates the influence of sea ice and SST variations in the BKS on the interannual fluctuations of rainfall within the Yangtze River basin. We show that enhanced Mei-yu rainfall in June instead of

summer mean over the Yangtze River basin is associated with BKS sea ice loss and SST increase.

## 2 Datasets and methodology

We use monthly precipitation data from the Climate Prediction Center Merged Analysis of Precipitation (CMAP) and the Climatic Research Unit Time Series (CRU TS) , monthly sea ice concentration (SIC) and SST data both from the Hadley Centre, and the European Centre for Medium-

range Weather Forecasts ERA5 reanalysis dataset, which includes monthly surface air temperature (SAT), geopotential height (GPH), sea level pressure (SLP), surface net thermal radiation, surface net solar



radiation, surface sensible heat flux, surface latent heat flux, specific humidity, zonal and meridional winds, and vertical velocity.

To test the robustness of the results, we employed the NCAR Community Atmosphere Model Version 5 (CAM5; Neale et al., 2012) to examine the potential influence of observed changes in BKS sea ice and SST on summer precipitation in East Asia. The CAM5 was run with a global horizontal resolution of about 2 degrees (F19_F19) and 30 levels in the vertical. The atmospheric model in each ensemble member was forced by monthly sea ice and SST data from the Hadley SST and sea ice datasets, covering the observed BKS region (65°–80° N, 0°–80° E) from 1982 to 2021. In other oceanic regions, SST and SIC forcings exhibited climatological averages characterized by a seasonal cycle. Meanwhile, other forcings, including the concentration of greenhouse gases, aerosols, and solar radiation, are all set to fixed values during the entire integration period. The ensemble average was calculated from the simulation results of 20 ensemble members to determine the collective model response to the observed variations in BKS SIC and SST.

The climatological mean is defined as the average of the data from 1982 to 2021. To mitigate the effects of global warming, a method for removing linear trends was employed before the analysis. The influence of May SST in the western North Atlantic was eliminated before analysis as suggested by Zheng and Wang (2021), which does not change the main conclusion of the present study. The study describes the atmospheric wave activity using the wave activity flux (WAF) as derived by Takaya and Nakamura (2001).

## 3 Results

### 3.1 Dominant features and relationships between BKS sea ice/SST and rainfall in the Yangtze River basin

Figures 1a and 1b show the distribution of correlation coefficients between June rainfall in the Yangtze River basin and BKS SIC/SST. Since the 1980s, prominent fluctuations in SIC have been documented in the BKS and coastal areas, exhibiting relatively high standard deviations (0.25 over the Kara Sea). There is a significant negative correlation between the June precipitation in the Yangtze River





basin and sea ice in the Kara Sea (r < -0.4) and a positive correlation with SST in both the Barents Sea and the Kara Sea (r > 0.4). A correlation of 0.27 reaches the 90% significance level based on a student *t*-test. The significant correlations between BKS SIC/SST and June rainfall in the Yangtze River basin indicate that the melting of sea ice and the resulting changes in SST in the BKS may have an impact on June precipitation in East Asia on the interannual timescale.

To measure the interannual variations in sea ice and SST, SIC and SST indices are established over the BKS region (69°–74° N, 45°–69° E). Over the last four decades, there has been a consistent decrease (-0.37 per decade) in SIC and a noteworthy warming trend (0.36℃ per decade) in SST, which are highly correlated with each other on the interannual timescale at -0.96. The reduction of sea ice results in a larger expanse of open water being exposed to sunlight than normal state, which has a lower albedo, consequently causing greater absorption of solar radiation and an increase in SST. The positive SST anomalies contribute to the acceleration of sea ice melting, leading to the establishment of a positive feedback loop (Kellogg, 1975; Curry et al., 1995; Screen and Simmonds, 2012).



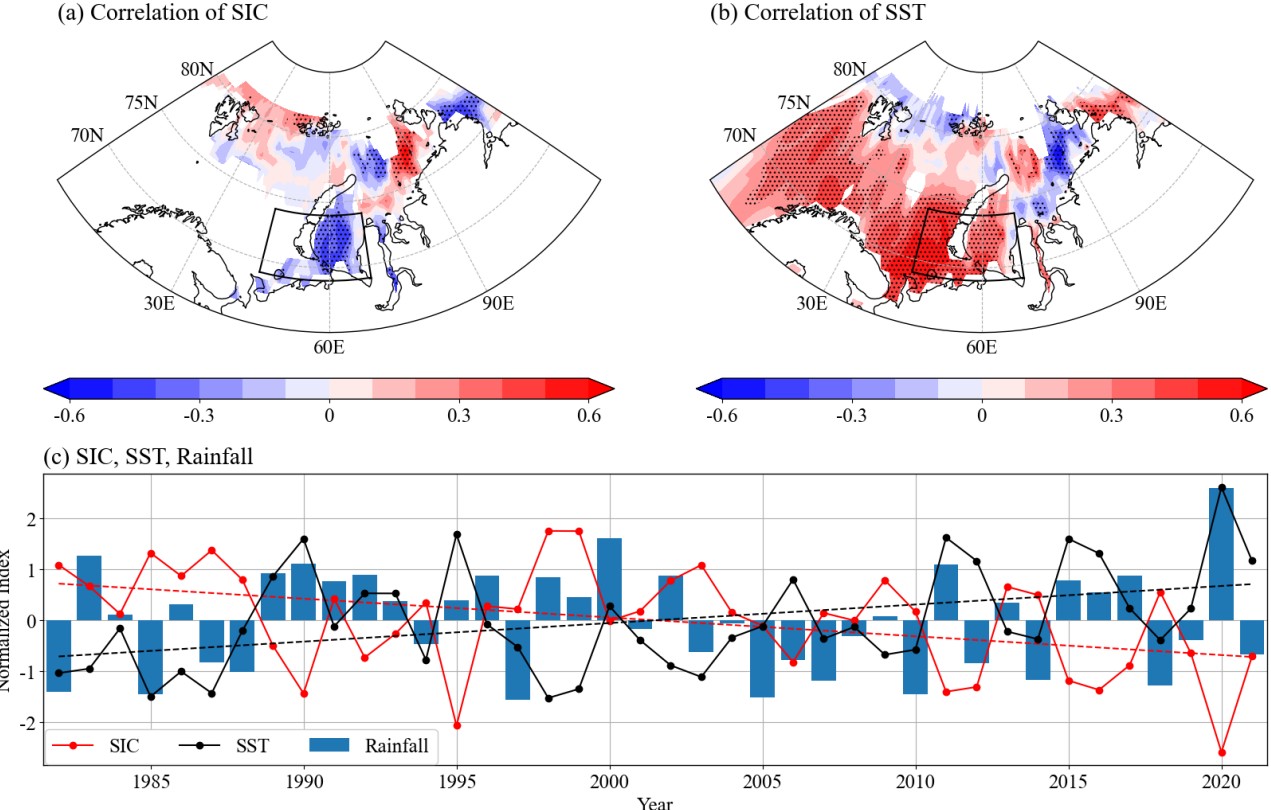

**Figure 1: Distribution of correlation coefficients between the June rainfall index over the Yangtze River basin (27.5°–34.7° N, 103°–122° E, Fig. 2a) and (a) SIC anomalies (omitted STD below 0.11) and (b) SST anomalies (omitted STD below 0.36 °C) in the BKS during 1982–2021. The areas significant at the 90% confidence level are dotted. (c) Normalized time series of the regionally averaged SIC/SST in the BKS (69°–74° N, 45°–69° E, black boxes) and rainfall in the Yangtze River basin. The dashed lines denote the linear trends.**

## 3.2 The mechanism of BKS sea ice/SST affecting rainfall in the Yangtze River basin

We use CMAP and CRU precipitation data to verify and ensure the consistency and reliability of our results. Figure 2a depicts the distribution of correlation coefficients between BKS SST and precipitation in June. In the Yangtze River basin, a positive correlation between SST and precipitation has been observed. This indicates that an increase in June SST in the BKS region corresponds with increased precipitation in the Yangtze River basin, supporting the findings shown in Fig. 1. Associated with an increase in Mei-yu rainfall, convective activities are depressed over the South China Sea and NWP region (Fig. 2b), while the low-level circulation anomalies show an AAC pattern and high SLP (Fig.




3b) anomaly. These patterns are largely consistent with summer mean anomalies during post-El Niño summer, known as the Indo–Western Pacific Ocean capacitor (IPOC) mode (Xie et al., 2016). Indeed, the correlation between the June BKS SST index and the preceding DJF Niño-3.4 index amounts to -0.02. The above regressions remain unchanged after removing the ENSO effect in the preceding winter. In 140 addition, there are positive rainfall anomalies over northeast Asia with a low-level anomalous cyclonic circulation anomaly pattern (Zhang et al., 2021).

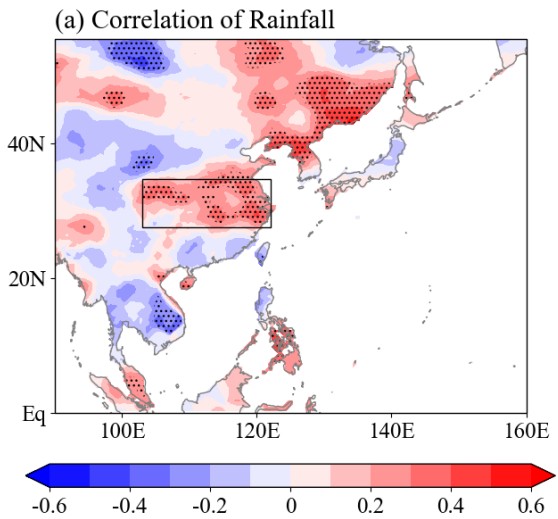
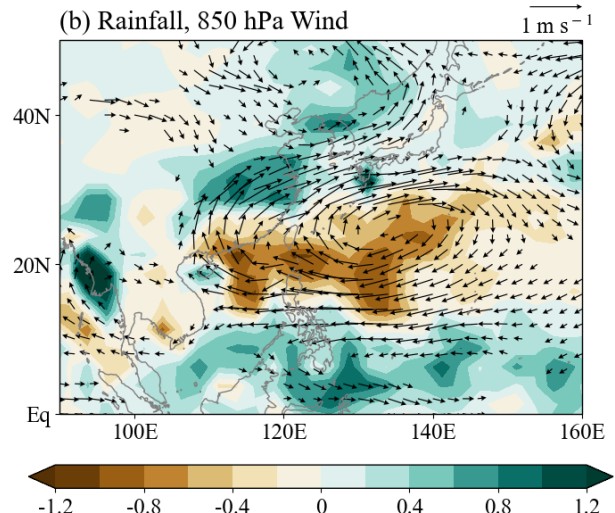

**Figure 2: (a) Correlation map of East Asia rainfall with concurrent BKS SST index in June during 1982–2021. The areas significant at the 90% confidence level are dotted. (b) Anomalies of rainfall (shading, unit: mm day⁻¹) and 850 hPa wind (vector, unit: m s⁻¹, omitted below 0.15 m s⁻¹) regressed onto the BKS SST index in June.**

Figure 3 depicts the regressed atmospheric circulation and temperature anomalies against the BKS SST index. A Rossby wave pattern, characterized by a sequence of positive, negative, and positive GPH anomalies, emerges from the Arctic to East Asia at 200 hPa (Fig. 3a). This atmospheric pattern exhibits 150 an equivalent barotropic vertical structure, extending from 200 hPa to the surface (Figs. 3a, 3b). This is accompanied by the propagation of WAF (Fig. 3c), signifying a significant influence of atmospheric activity on rainfall in East Asia. Specifically, a pronounced positive GPH anomaly is present over the BKS and northern Siberia. According to Kim et al. (2022), Siberia frequently encounters elevated temperatures during the summer season, and heatwaves linked to high-pressure heat domes contribute to



the development of regional high-pressure systems, aligning with the findings of our study. The anomalous anticyclonic circulation linked to high pressure (Fig. 3b) facilitates the northward transport of warm air (Fig. 5a), causing the sea ice melting and positive SST anomalies in the BKS (Fig. 3d). The reduced sea ice coverage increases open water and decreases the sea ice albedo, and reflects less solar radiation (Fig. 4b) and enhances the ocean warming (Fig. 3d) as positive ice-albedo feedback.

Furthermore, the melting of snow cover in the northern Siberia (Fig. 4d) contributes to the increase in the low-level water vapor (Fig. 5b), which effectively traps outward longwave radiation and induces upward sensible and latent heat fluxes in the northern Siberia region (Figs. 4c, 4d), maintaining the high-pressure system (Figs. 3a, 3b) as a positive loop. However, the intensified net heat flux (Fig. 4a) from the atmosphere to the ocean in the BKS region is not a direct cause of sea ice melting. Rather, it is a

consequence of the decreased sea ice concentration in this region. The melting sea ice reflects less solar radiation from the ocean to the atmosphere, thereby increasing the downward net heat flux (Fig. 4a), which is mostly contributed by the incoming shortwave radiation (Fig. 4b).

To further elucidate the propagation characteristics of the Rossby wave train, an in-depth analysis was conducted on the WAF profiles along the Rossby wave path (Fig. 3c). The overlaid arrows represent

the horizontal and vertical components of the WAF. This illustration reveals a pronounced transmission of Rossby waves from the BKS region to East Asia within the mid-to-upper atmosphere. Simultaneously, the thermal high-pressure induced by surface heat anomalies facilitates the upward propagation of WAF, thereby enhancing the energy of Rossby waves. The GPH anomalies manifest a distinct quasi-barotropic structure characterized by '+-+' anomalies along the propagation pathway, which contributes to the

dissipation of WAF, ultimately reaching East Asia. The negative (positive) GPH anomaly center over the Baikal Lake and northeast Asia (NWP region) is accompanied by low-level anomalous cyclone (anticyclone). The southwesterly of the NWP AAC and northwesterly of the anomalous cyclone over the northeast Asia converge in the Yangtze River basin, contributing to the enhanced rainfall there (Fig. 2). These observations offer additional evidence that the Arctic domain, particularly the BKS, serves as a

significant source of variability in East Asian precipitation. The relay of wave energy could be a fundamental physical mechanism through which sea ice/SST influences precipitation patterns in East Asia.



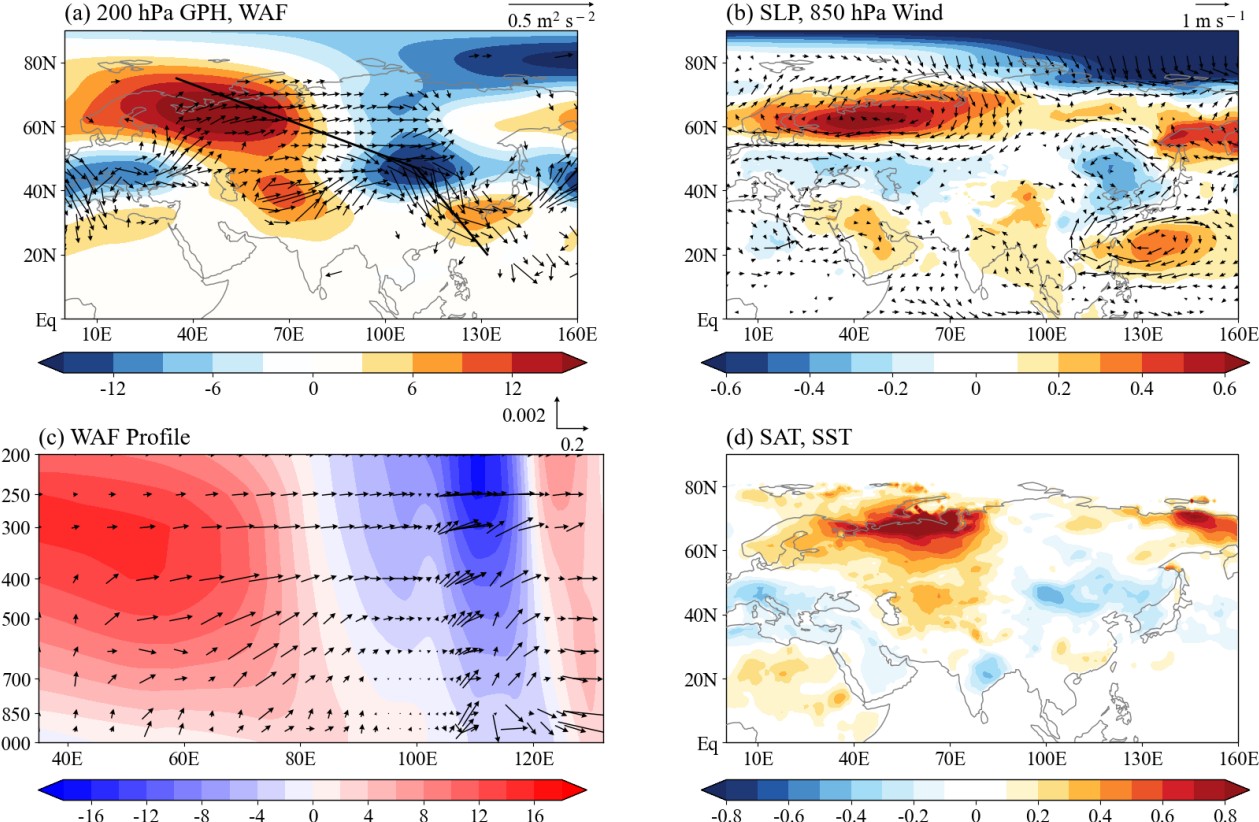

**Figure 3: Anomalies of (a) 200 hPa GPH (shading, unit: gpm) and horizontal WAF (vector, unit: m² s⁻², omitted below 0.05 m² s⁻²), (b) SLP (shading, unit: mbar) and 850 hPa wind (vector, unit: m s⁻¹, omitted below 0.1 m s⁻¹), (c) vertical–horizontal cross-section averaged within (20°–75° N, 35°–132° E) for WAF (vector, unit: m² s⁻²) and GPH (shading, unit: gpm), (d) SAT (shading, unit: °C, on land) and SST (shading, unit: °C, on ocean) regressed onto the BKS SST index in June.**



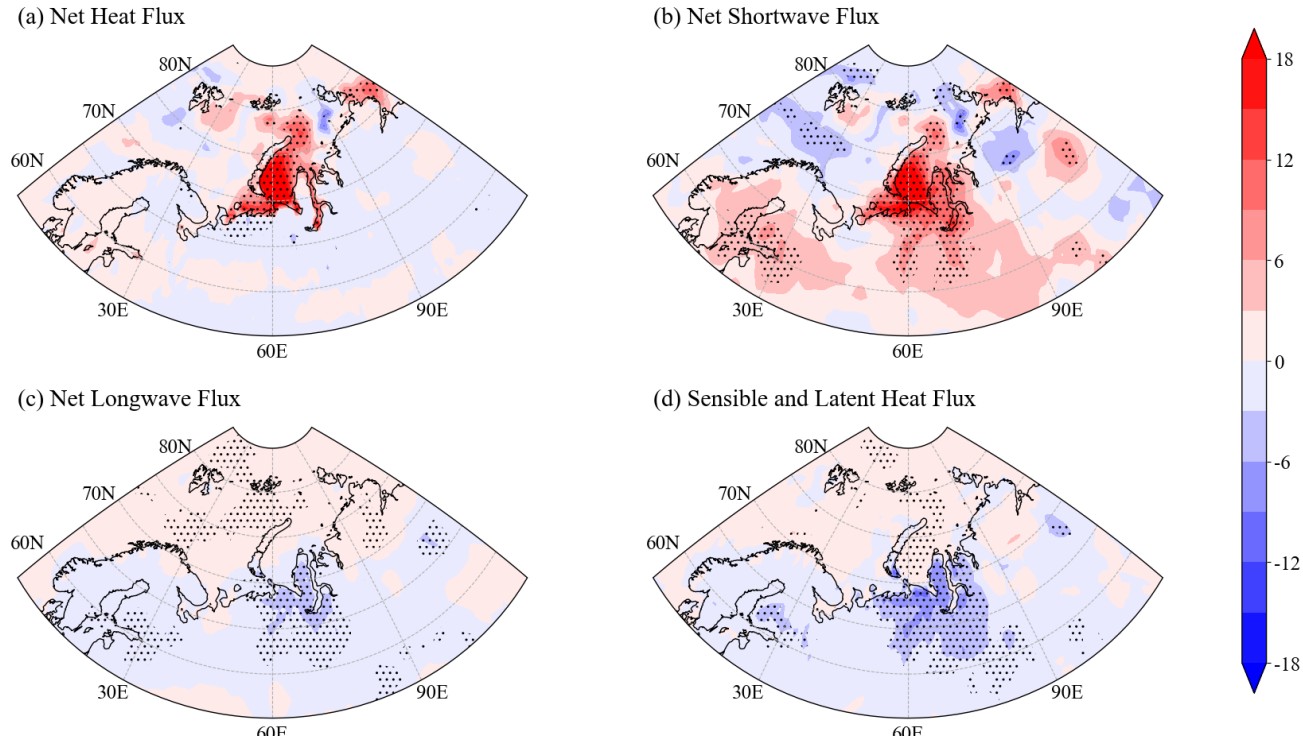

**Figure 4: Anomalies of (a) net heat flux, (b) net shortwave flux, (c) net longwave flux, (d) sensible heat flux and latent heat flux (shading, unit: W m⁻²) regressed onto the BKS SST index in June. The areas significant at the 90% confidence level are dotted.**

The other high-pressure center of the quasi-barotropic Rossby wave is located over the East Asia and NWP region, resulting in the enhancement of NWP subtropical high and depressed convective activities (Figs. 2b, 3a, 3b). The accompanying 850 hPa wind anomalies show an anomalous anticyclone pattern, which can transport increased warm water vapor to the Mei-yu front (Fig. 5c). The low-pressure center of the quasi-barotropic Rossby wave located over the Baikal Lake is accompanied by low-level anomalous cyclone. The southwesterly of the NWP AAC and northwesterly of the anomalous cyclone over the northeast Asia converge in the Yangtze River basin, contributing to the enhanced rainfall there (Fig. 2b). In addition, coupled with localized upward vertical velocities (Fig. 5c), this leads to atmospheric uplift, convective development, and the observed positive precipitation anomalies (Fig. 2b). At the 500 hPa level, the intensification of the westerly jet acts as a conduit for warm advection toward the precipitation zones, as shown in Fig. 5d, promoting convection and leading to the observed increase in precipitation within the region as feedback between latent heating and circulation anomalies (Lu and Lin,



2009; Sampe and Xie, 2010; Kosaka et al., 2011; Zhou et al., 2021). All of the three above processes are contributing to increased rainfall in the Yangtze River basin during June.

**Figure 5: Anomalies of (a) 850 hPa horizontal temperature advection (shading, unit: $10^{-7}$ K s$^{-1}$) and 850 hPa wind (vector, unit: m s$^{-1}$), (b) 850 hPa specific humidity (shading, unit: kg kg$^{-1}$) with significant areas at 90%**
**confidence dotted, (c) moisture flux (vector, unit: kg m$^{-1}$ s$^{-1}$, omitted below 5 kg m$^{-1}$ s$^{-1}$) and 500 hPa omega (shading, unit: $10^{-3}$ Pa s$^{-1}$), (d) 500 hPa horizontal temperature advection (shading, unit: $10^{-7}$ K s$^{-1}$) and 500 hPa wind (vector, unit: m s$^{-1}$, omitted below 0.15 m s$^{-1}$) regressed onto the BKS SST index in June.**

The investigation of teleconnections between Arctic climate anomalies and East Asian meteorological patterns employs Empirical Orthogonal Function (EOF) analysis, focusing on GPH
deviations within the wave train corridor spanning (30°–80° N, 40°–140° E). Figure 6a reveals an elevated GPH anomaly centered over the BKS and the northwestern Eurasian sector, consistent with observed atmospheric configurations. The designated horizontal WAF originates from the BKS and extends



southeastward from northern Siberia, dynamically reinforced by regional high-pressure systems. This flux intensifies near the Baikal Lake and maintains its trajectory toward the Yangtze River basin, illustrating the wave train's role in modulating the East Asian climate. A noticeable westward shift in the modeled high-pressure center over the BKS than the observations may be linked to prominent North Atlantic SST anomalies, a deviation from climatological data requiring further investigation. In addition, the position of both low-pressure center near Baikal Lake (Fig. 6a) and positive Mei-yu rain belt anomalies (Fig. 6b) are northward shifted than the observations, this is due to inherent biases in CAM5 model with a northward displacement of the mean westerly jet and climatological Mei-yu rain belt over the Yangtze River basin in June and July (not shown), as consistent with CAM6 model (Zhou et al., 2021).

Overall, the findings suggest that significant June anomalies in sea ice and SST within the BKS trigger anomalous local surface heating. This thermal disturbance serves as a genesis point for teleconnection patterns, propagating across the Eurasian landmass and significantly impacting the East Asian climate system.

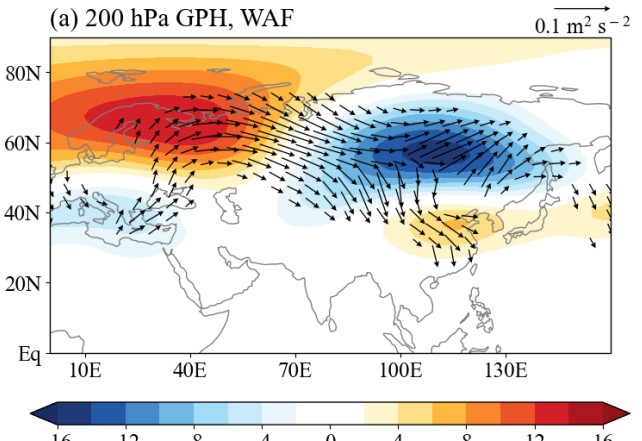
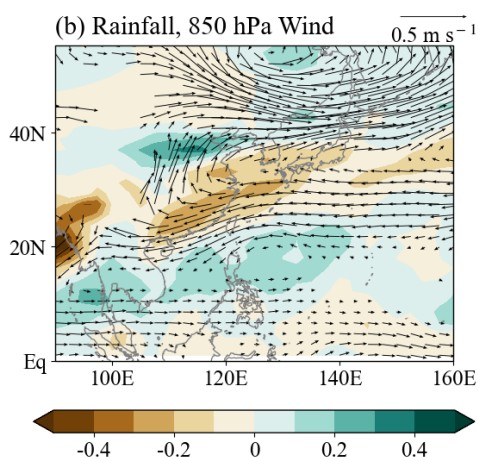

**Figure 6: Anomalies of (a) 200 hPa GPH (shading, unit: gpm) and horizontal WAF (vector, unit: m² s ⁻², omitted below 0.02 m² s ⁻²), (b) rainfall (shading, unit: mm day⁻¹) and 850 hPa wind (vector, unit: m s⁻¹, omitted below 0.03 m s⁻¹) regressed onto the second principle component (PC2) from the EOF analysis during 1982–2021.**





## 4 Conclusion and discussion

The present study investigates the significant influence of the BKS on East Asian precipitation, utilizing both observational data and numerical experiments to delineate the underlying mechanisms. The findings highlight the complex interplay between June sea ice and SST anomalies in the BKS and concurrent variations in precipitation across East Asia, as mediated by complex atmospheric circulation patterns.

The analysis reveals a distinctive '+-+' quasi-barotropic structure of GPH field in the mid-to-upper level, indicating the propagation of Rossby wave trains across the Eurasian continent. Notably, the BKS and northern Siberia emerge as regions of pronounced high-pressure anomalies, facilitating anticyclonic circulations. These circulation anomalies induce atmospheric warm advection and enhance shortwave radiation absorption, contributing to ocean warming, sea ice melting, and subsequent SST increases. The amplification of shortwave radiation absorption by the ocean not only enhances local water vapor content but also escalates downward longwave radiation, latent, and sensible heat fluxes, exacerbating the regional temperature rise. The reduced sea ice reflects less solar radiation and enhances the ocean warming following the ice-albedo feedback, which together with melting snow cover in the northern Siberian region contribute to the high-pressure system through thermal processes. A positive feedback loop between oceanic and atmospheric dynamics cultivates the development of a robust heat dome, extending from the lower to the upper troposphere. Furthermore, the low (high) pressure anomaly center over the Baikal Lake and northeast Asia (NWP region) is accompanied by low-level anomalous cyclone (anticyclone). The southwesterly of the NWP AAC and northwesterly of the anomalous cyclone over the northeast Asia converge in the Yangtze River basin, aligning with the intensification of warm advection by intensified westerly jet, contributing to the enhanced Mei-yu rainfall over the Yangtze River basin.

Our exploration into the sea ice/SST in the BKS not only illuminates the rapid and profound implications of global warming on the Arctic but also introduces new dimensions to understanding precipitation variations in East Asia. As the Arctic continues to experience warming, its influence on East Asian precipitation patterns is likely to grow both in significance and complexity. Therefore, further research and dialogue on the changing Arctic environment and its impact on East Asian precipitation are

crucial. Enhanced observational efforts and comprehensive analyses are imperative to refine our understanding of these regional climate patterns. Such advancements will be instrumental in improving predictions of future climate changes, and the evolution of the Mei-yu band, and will underpin more effective disaster management and climate adaptation strategies.

## Data Availability

SST and SIC are obtained from the Met Office Hadley Centre (https://www.metoffice.gov.uk/hadobs/hadisst/index.html). Precipitation datasets are from NOAA (https://www.cpc.ncep.noaa.gov/products/global_precip/html/wpage.cmap.shtml) and the University of East Anglia Climatic Research Unit (https://crudata.uea.ac.uk/cru/data/hrg/). The ERA5 datasets are available online at https://cds.climate.copernicus.eu/cdsapp#!/search?type=dataset. Numerical experiments data have been deposited in GitHub (https://github.com/Tianli-Xie/numerical-experiments-for-precipitation-in-the-Yangtze-River-basin).

## Author contributions

TX analyzed the observational data and numerical experiment results. ZQZ conceived the study and conducted the numerical simulations. ZQZ and TX prepared the manuscript. All authors actively discussed the results and made significant contributions to the writing of the final manuscript.

## Competing interests

The contact author has declared that none of the authors has any competing interests.



## Disclaimer

Publisher's note: Copernicus Publications remains neutral with regard to jurisdictional claims made in the text, published maps, institutional affiliations, or any other geographical representation in this paper. While Copernicus Publications makes every effort to include appropriate place names, the final responsibility lies with the authors.

## Acknowledgments

We wish to thank Shuoyi Ding, Xuanwen Zhang, and Ruonan Zhang for useful discussions. This work is supported by the National Natural Science Foundation of China (42288101, 42175025) and the National Key Research and Development Program of China (2023YFF0806700).

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
