# Peer review of "Sea ice reduction in the Barents-Kara Sea enhances June precipitation in the Yangtze River basin"

_EGUsphere, 2024_

## Author Comment (AC1)

**Reply to Reviewer #1:**

[Figure]

Figure R1: Climatological (a) SIC and (c) SST. Standard deviation of (b) SIC and (d) SST anomalies (shading).

[Figure]

Figure R2: Distribution of correlation coefficients between the June SIC index over the Kara Sea (69°-74° N, 55°–69° E, black box) and (a) SIC anomalies, and between the June SST index and (b) SST anomalies during 1982–2021. The areas significant at the 90% confidence level are dotted.

[Figure]

Figure R3: First leading SVD mode between geopotential height at 250 hPa (left) and SST (right) fields during June 1982-2021.

---

## Author Comment (AC3)

**Reply to Reviewer #3:**

[Figure]

**Fig. R1** Same as Fig. 2a, but for July.

---

## Author Comment (AC4)

**Reply to Reviewer #3:**

[Figure]

**Fig. R1** Same as Fig. 2a, but for July.

---

## Author Response (AR1)

**Reply to Reviewer #1:**

*This study examines the relationship between June sea surface temperature (SST) and sea ice in the Barents-Kara Sea (BKS) and rainfall variability in the Yangtze River basin from 1982 to 2021. It finds that the observed decline in sea ice and rise in SST over the BKS region correlate with increased precipitation in the Yangtze River basin on an interannual timescale. The research identifies a barotropic Rossby wave train triggered by BKS thermal forcing, which propagates southeastward, contributing to enhanced rainfall through the strengthening of the subtropical high over the northwest Pacific region. These results have significant implications for summer rainfall predictions in East Asia. The paper is well organized and I have some comments and questions.*

Thank you for the comments. We have implemented all the suggestions as detailed below. The original comments are quoted in Italic.

*1. Figure 1a displays regions of white within the SIC correlation coefficient map. These areas could either represent absence of sea ice in June or result from low standard deviation values. Given the potential for minimal or no sea ice presence, it would be helpful if the authors elucidated the rationale behind extending the study area beyond the Kara Sea. A more comprehensive explanation is warranted. In addition, delete either one colorbar in Fig.1a and Fig.1b since they are same.*

**Response:** Following the suggestion, we have reviewed the SIC climatology and variance for June, which shows that the southern Barents Sea exhibits minimal sea ice coverage (**Fig. R1a**) and relatively low standard deviation (<0.11, **Fig. R1b**). However, in this region, SST shows significant interannual variability (>0.36, **Fig. R1d**). Therefore, the selection of our study area is based on the interannual variability of both SIC and SST. Additionally, the SIC and SST variations in the Kara Sea are consistent with those in the Barents Sea during the same period (**Fig. R2**). Based on these considerations, we have chosen to include both the Barents and Kara Seas as the study area. Furthermore, the figure has been revised as suggested. (**Fig.1**)

[Figure]

Figure R1: Climatological (a) SIC and (c) SST. Standard deviation of (b) SIC and (d) SST anomalies (shading).

[Figure]

Figure R2: Distribution of correlation coefficients between the June SIC index over the Kara Sea (69°-74°N, 55°–69°E, black box) and (a) SIC anomalies, and between the June SST index and (b) SST anomalies during 1982–2021. The areas significant at the 90% confidence level are dotted.

*2. The manuscript provides an extensive description of radiation flux; however, the definitions of the directional components within this context lacks clarity. It is advisable for the authors to incorporate precise definitions for these components to enhance understanding.*

**Response:** Modified as suggested. (**Lines 100-101**)

*3. Figure 6 presents the results from the model simulations, utilizing the second principal component (PC2) of the EOF analysis, which indicates the presence of a wave-like pattern. This prompts two critical inquiries: Firstly, which EOF mode in the observations corresponds to the '+-+' pattern?*

*Secondly, do the model simulations align with the observations?*

**Response:** Following the suggestion, we conducted a Singular Value Decomposition (SVD) analysis (Fig. R3). The leading SVD mode was consistent with the '+-+' pattern shown in **Fig.3a**, which explains 53% of the total covariance. Furthermore, the model simulations **(Fig.6a)** are consistent with the observational results. We have included this discussion in the revised manuscript. **(Line 227-229)**

[Figure]

[Figure]

Figure R3: Anomalies of (a) 200 hPa GPH (shading, unit: gpm) (b) SST (shading, unit: ℃, on ocean) and surface air temperature (SAT, shading, unit: ℃, on land) regressed onto the SVD mode time coefficients for SST during 1982–2021. The areas significant at the 90% confidence level are dotted.

*4. In Figure 5c, the units of the arrows should be specified as kg m⁻¹ s⁻¹. It is advisable for the authors to amend this notation in the figure accordingly.*

**Response:** Revised as suggested. **(Fig.4)**

*5. The manuscript states, "However, the intensified net heat flux from the atmosphere to the ocean in the BKS region is not a direct cause of sea ice melting, but rather a consequence of decreased sea ice concentration." It is recommended that the authors further elucidate the causal relationship between these phenomena.*

**Response:** Modified as suggested. The melting of sea ice results in an enhanced net heat flux from the atmosphere to the ocean in the BKS, which is consistent with the ice-albedo feedback mechanism (Kellogg, 1975; Curry et al., 1995; Screen and Simmonds, 2012). We have included this discussion in the revised manuscript. **(Lines 174-179)**

Reference:
Curry, J. A., Schramm, J. L., and Ebert, E. E.: Sea ice-albedo climate feedback mechanism, Journal of Climate, 8, 240-247, 1995.
Kellogg, W. W.: Climatic feedback mechanisms involving the polar regions, Climate of the Arctic, 111-116, 1975.

Screen, J. A. and Simmonds, I.: Declining summer snowfall in the Arctic: Causes, impacts and feedbacks, Climate dynamics, 38, 2243-2256, 2012.

**Reply to Reviewer #2:**

*This study investigates the potential relation in variability between the Barents-Kara seaice and the June rainfall in Yangtze River Basin. They show that the observed decrease in BKS sea ice and the corresponding increase in SST during June aligns with enhanced precipitation in the Yangtze River at interannual timescale. The BKS thermal forcing induces a barotropic Rossby wave train that propagates southeastward to the Northwest Pacific (NWP), leading to an anomalous anticyclone that enhances rainfall in the Yangtze River Basin. The results are worth publishing. I recommend minor revision with the following comments:*

Thank you for the comments. We have implemented all the suggestions as detailed below. The original comments are quoted in Italic.

*1. Fig. 1: Are the correlations in Fig. 1a and 1b computed based on detrended data, since we are talking about correlations at interannual time scale? What is the correlation between the time series of Rainfall and SIC/SST shown in Fig. 1c? I wonder if these correlations still hold for July. If not, what causes the differences between June and July?*

**Response:** Yes, the correlations in Fig.1a and 1b were computed using detrended data as introduced in the "Datasets and Methodology" section of the manuscript. Following your suggestion, we have calculated the correlation between the time series of rainfall and SIC/SST shown in **Fig. 1c**, which are -0.37/0.44 during 1982-2021, both above the 0.1 significance level. For July, the correlations between BKS SST and rainfall in the Yangtze River Basin become insignificant (**Fig. R4**), which might be due to the northward movement of the mean rain belt. Additionally, according to Wu et al. (2023), the anomalous melting of Arctic sea ice primarily affects East Asian summer (JJA) precipitation in middle and high latitudes regions, with minimal impacts on rainfall in the Yangtze River basin. As a result, August shows a similar relationship as in July, which might contribute to the seasonal mean anomalies being insignificant as in Wu et al. 2023. In contrast, our study highlights that changes in BKS SIC/SST in June significantly influence rainfall in the Yangtze River Basin as one of the major findings in the present study. The reasons for the reduced impact in July and August need further investigation. We have included this discussion in the revised manuscript. **(Lines 121-124, Lines 145-150)**

Reference:
Wu, B., Li, Z., Zhang, X., Sha, Y., Duan, X., Pang, X., and Ding, S.: Has Arctic sea ice loss affected summer precipitation in North China?, International Journal of Climatology, 2023.

[Figure]

Figure R4: Same as Fig. 2a, but for July.

*2.Fig.3c, 5a, 5b: Fig. 3c: no unit for y axis pressure cooredinate.*
*Fig. 5a is mentioned before Fig. 4 in the main text.*
*Fig. 5b: should the unit of specific humidity be g kg-1, considering the value can be 1?*

**Response:** Modified as suggested. (**Fig.3c and Fig.4**)

*3. Around Line 160: I understand how melting sea-ice warms the surface, but how does the warm surface leads to the high-pressure system? An explanation is needed for this step.*

**Response:** Clarified as suggested. The melting sea-ice induced low-level warm can enhance the whole high-pressure system as which exhibiting a quasi-barotropic structure with upward wave activities (**Fig.3c**), as a result, the intensified high-pressure system leads to increased shortwave radiation (**Fig. 5b**) and further snow and sea ice reductions, as a positive feedback loop. We have added the above description in the revised manuscript. (**Line 171-174**)

*4. Line 175-178 versus Line193-196: I am a little confused. Are these two sentences describing the same information? If not, please explain more specifically. If so, the authors may consider remove such redundancy.*

**Response:** Modified as suggested. (**Line 185-188 and Line 206-208**)

*5. The authors mentioned the CAM5 simulations in the methodology. However, it is unclear to me where they present the CAM5 result. It should be stated more clearly in the main text and figure captions which is from observations, and which is from CAM5 simulations.*

**Response:** Modified as suggested. **Fig. 6** is the only figure from CAM5 simulations. (**Line 225 and Fig.6**)

**List of Revisions (Based on Track-changes file.pdf)**

1. Page 4 Line 100, "In the following figures, positive values indicate downward radiative fluxes, while negative values represent upward radiative fluxes." was added.

2. Page 5 Line 121, "From 1982 to 2021, the correlation coefficients between June rainfall in the Yangtze River basin and BKS SIC/SST reach -0.37 and 0.44, both are significant at the 90% confidence level, indicating the influence of BKS SIC/SST variations on East Asian summer precipitation." was added.

3. Figure 1 was replaced with an updated version.

4. Page 8 Line 147, "However, in July and August, correlations between BKS SIC/SST and Yangtze River rainfall become insignificant (not shown), possibly due to the northward shift of the mean rain belt. This intraseasonal difference is consistent with findings by Wu et al. (2023) that the anomalous melting of Arctic sea ice primarily affects East Asian summer seasonal mean precipitation in middle and high latitudes regions, with minimal impacts on rainfall in the Yangtze River basin. Further investigation is needed to understand these monthly differences." was added.

5. Page 9 Line 169, the statements of "Fig. 5a" were corrected as "Fig. 4a".

6. Page 9 Line 171, the statements of "Fig. 4b" were corrected as "Fig. 5b".

7. Page 9 Line 172, the statements of "Fig. 4d" were corrected as "Fig. 5d".

8. Page 9 Line 173, the statements of "Fig. 5b" were corrected as "Fig. 4b".

9. Page 9 Line 174, the statements of "Figs. 4c, 4d" were corrected as "Figs. 5c, 5d".

10. Page 9 Line 174, the statements of ", maintaining the high-pressure system (Figs. 3a, 3b) as a positive loop." were corrected as "This enhances the high-pressure system, which exhibits a quasi-barotropic structure with upward wave activities (Fig. 3c). As the high-pressure system (Figs. 3a, 3b) intensifies, it increases shortwave radiation (Fig. 5b), accelerating snow melt, thus reinforcing the high-pressure system in a positive feedback loop.".

11. Page 9 Line 179, the statements of "Fig. 4a" were corrected as "Fig. 5a".

12. Page 9 Line 180, the citation of "(Kellogg, 1975; Curry et al., 1995; Screen and Simmonds, 2012)" was added

13. Page 9 Line 182, the statements of "Fig. 4a" were corrected as "Fig. 5a".

14. Page 9 Line 183, the statements of "Fig. 4b" were corrected as "Fig. 5b".

15. Page 10 Line 190, ", with the negative (positive) GPH anomaly center located over the Baikal Lake and northeast Asia (NWP region), accompanied by low-level anomalous cyclones (anticyclones, )." was added.

16. Page 10 Line 192, the statements of ", which" were corrected as "This quasi-barotropic structure".

17. Page 10 Line 193, "The negative (positive) GPH anomaly center over the Baikal Lake and northeast Asia (NWP region) is accompanied by low-level anomalous cyclone (anticyclone). The southwesterly of the NWP AAC and northwesterly of the anomalous cyclone over the northeast Asia converge in the Yangtze River basin, contributing to the enhanced rainfall there (Fig. 2)." was deleted.

18. Figure 3 was replaced with an updated version.

19. Figure 4 and the caption was replaced with an updated version.

20. Page 14 Line 219, the statements of "The other high-pressure center of the quasi-barotropic

Rossby wave is located over the East Asia and NWP region, resulting in the enhancement of NWP subtropical high and depressed convective activities (Figs. 2b, 3a, 3b).＂ were corrected as ＂In East Asia, the other high-pressure center of the quasi-barotropic Rossby wave influences the NWP region. This high-pressure system strengthens the NWP subtropical high and reduces local convective activities (Figs. 2b, 3a, 3b).＂.

21. Page 14 Line 224, the statements of ＂Fig. 5c＂ were corrected as ＂Fig. 4c＂.
22. Page 15 Line 228, the statements of ＂Fig. 5c＂ were corrected as ＂Fig. 4c＂.
23. Page 15 Line 231, the statements of ＂Fig. 5d＂ were corrected as ＂Fig. 4d＂.
24. Figure 5 and the caption was replaced with an updated version.
25. Page 16 Line 247, ＂using the CAM5 model＂ was added.
26. Page 16 Line250, "which is the leading Singular Value Decomposition (SVD) mode between GPH at 250hPa and SST fields during 1982-2021." was added.
27. The caption of Fig.6 was replaced with an updated version.